# Assessment of quality of antenatal care services in public sector facilities in India

Rakhi Dandona ![ORCID] ,[1,2] Moutushi Majumder,[1] Md Akbar,[1] Debarshi Bhattacharya,[3] Priya Nanda,[3] G Anil Kumar,[1] Lalit Dandona[1,2]

¹Public Health Foundation of India, Gurugram, Haryana, India
²Department of Health Metrics Sciences, University of Washington, Seattle, Washington, USA
³Bill & Melinda Gates Foundation India, New Delhi, India

**Correspondence to**
Dr Rakhi Dandona;
rakhi.dandona@phfi.org

## ABSTRACT

**Objectives** We undertook assessment of quality of antenatal care (ANC) services in public sector facilities in the Indian state of Bihar state delivered under the national ANC programme (Pradhan Mantri Surakshit Matritva Abhiyan, PMSMA).

**Setting** Three community health centres and one subdistrict hospital each in two randomly selected districts of Bihar.

**Participants** Pregnant women who sought ANC services under PMSMA irrespective of the pregnancy trimester.

**Primary and secondary measures** Quality ANC services were considered if a woman received all of these services in that visit—weight, blood pressure and abdomen check, urine and blood sample taken, and were given iron and folic acid and calcium tablets. The process of ANC service provision was documented.

**Results** Eight hundred and fourteen (94.5% participation) women participated. Coverage of quality ANC services was 30.4% (95% CI 27.3% to 33.7%) irrespective of pregnancy trimester, and was similar in both districts and ranged 3%–83.1% across the facilities. Quality ANC service coverage was significantly lower for women in the first trimester of pregnancy (6.8%, 95% CI 3.3% to 13.6%) as compared with those in the second (34.4%, 95% CI 29.9% to 39.1%) and third (32.9%, 95% CI 27.9% to 38.3%) trimester of pregnancy. Individually, the coverage of weight and blood pressure check-up, receipt of iron folic acid (IFA) and calcium tablets, and blood sample collection was >85%. The coverage of urine sample collection was 46.3% (95% CI 42.9% to 49.7%) and of abdomen check-up was 62% (95% CI 58.6% to 65.3%). Poor information sharing post check-up was done with the pregnant women. Varied implementation of ANC service provision was seen in the facilities as compared with the PMSMA guidelines, in particular with laboratory diagnostics and doctor consultation. Task shifting from doctors to ANMs was observed in all facilities.

**Conclusions** Grossly inadequate quality ANC services under the PMSMA needs urgent attention to improve maternal and neonatal health outcomes.

## STRENGTHS AND LIMITATIONS OF THIS STUDY

⇒ Data on the quality of antenatal care services received by pregnant women documented in public sector health facilities using exit survey.
⇒ Information provided to the pregnant women after the antenatal care check-up documented in exit survey.
⇒ Direct observations made for the process of antenatal care services offered in the public sector health facilities.
⇒ Study was limited to six public sector health facilities with no private sector assessment, which could limit generalisation of the findings.
⇒ Exclusion of 10% of women due to non-availability of a phone for contact may not have any significant impact on the findings.

## BACKGROUND

The inadequacy of and inequity in quality of antenatal care (ANC) services is increasingly receiving attention.[1–7] The key elements of ANC services captured under the facility surveys also indicate inadequate quality of care provision.[8] Concerns about the quality of ANC services have also been reported from India, primarily based on the retrospective data collected under the National Family Health Survey.[1 9–11] The ANC quality in these reports is captured retrospectively and does not offer nuanced understanding of what happens in a particular ANC visit with a health provider, thereby, limiting the actions needed to address the coverage-quality gap in ANC services.

To increase the coverage and quality of ANC services, the Government of India started the Pradhan Mantri Surakshit Matritva Abhiyan (PMSMA; Prime Minister's Safe Motherhood Programme).[12] PMSMA is based on the premise that if every pregnant woman in India is examined by a physician and appropriately investigated at least once during the PMSMA and then appropriately followed up, the process can result in reduction in the number of maternal and neonatal deaths in the country. With over 18 000 facilities providing PMSMA services, nearly 28.1 million pregnant women have been examined in India under this programme

since its inception in 2016.[12] However, no comprehensive assessment of ANC services under the PMSMA is available in the public domain. In this context, we undertook a health facility based assessment of ANC services offered on the PMSMA day in the Indian state of Bihar, one of the most populous Indian states.[13] The coverage of at least one ANC visit in the state is at 82% and that for 4 ANC visits at 25.2%,[14] with the neonatal mortality rate in 2017 at 23.4 per 1000 livebirths and maternal mortality ratio of 230.[15 16] We report on the quality of ANC services, information sharing with the pregnant women based on the ANC check-up, and deviations from the prescribed process of ANC service provision under PMSMA with the aim to identify the areas that needed attention for the PMSMA to achieve its stated goal.

## METHODS
### Study design

#### Selection of health facilities
A multistage sampling process was followed to select the public sector health facilities for this study. First, two districts from the state were sampled based on the latest available 3+ visits ANC coverage for Bihar for 2016 at the time of planning of this study.[17] All the 38 districts of the state were grouped into two strata as same/above and below the state median coverage for 3+ visits ANC visits (29.5%). We then randomly selected one district each from these two groups for this study—Gaya and Supaul. In each of the two sampled districts, three block-level community health centres and one subdistrict hospital (SDH) were purposively sampled from the public sector health facilities where detailed assessments were already being undertaken.

#### Selection of pregnant women
All pregnant women who reported for ANC services to the sampled health facilities, irrespective of the trimester of pregnancy, on the ninth day of the study month under the PMSMA scheme were considered. The only exclusion criterion was non-availability of a phone number for contact later for follow-up assessment. We aimed to recruit 200 pregnant women in each of the sampled facilities, giving a total sample of 1200 women across the six facilities.

#### Data collection
Data collection began in September 2019 in Gaya district but was then withheld in October and November 2019 due to floods and sociopolitical issues in the state, and was next two rounds were completed in December 2019 and January 2020. Data collection in Supaul district was started in February 2020 and could not be continued due to the COVID-19 pandemic lockdown. Hence, the data available for analysis were for three rounds in Gaya district and for one round of data collection in Supaul

district. The permission to conduct the study was sought from the health facility-in-charge.

#### Exit interviews
In each data collection round, exit interviews were conducted with the pregnant women. Interviewers trained in the study procedures contacted pregnant women after they registered at the outpatient department (OPD) registration for ANC services. Those who were able to provide a phone number were explained in brief about the purpose of the study and requested to wait after they had availed ANC services on that day. The team posted at the health facility kept track of these women through the various steps of ANC services, and recontacted them after they had availed the services. They were provided refreshments and explained the purpose of the study in detail. Those who consented were interviewed in a semiprivate place in the facility premises. The exit interviews documented sociodemographic information of the pregnant woman, details of the ANC services provided on the given PMSMA day, women's knowledge of the services received and what they were informed post the check-up.

The exit interview tool was developed in English and then translated into Hindi (local language), after which back translated into English to ensure accurate and relevant meaning and intent of the questions. Pilot testing of the tool was carried out and modifications made as necessary. The interviews were captured using the Computer-Assisted Personal Interviews software in handheld tablets. Data entered were scrutinised using the internal checks, and some portion of the 30% of all interviews were also attended by a supervisor for quality control purpose with prior permission taken from the respondent to do so.

#### Observation of the flow of ANC services
The interviewers placed in each facility for the exit interviews also observed the stepwise flow of ANC service provision followed for the pregnant women. The entire process followed on a given day was documented in a format specifically developed to track the process from the beginning till the end. The number and type of staff providing ANC services and the number of pregnant women registered for ANC services were documented.

#### Data analysis
We present the coverage of each ANC component by the pregnancy trimester, facility and district from the exit interviews. Whether pregnant women have received some or all components of a set of interventions as part of ANC at least once during pregnancy has been used to indicate quality of care.[18–20] For this analysis, we defined quality ANC service when a pregnant woman reported receiving all of the following ANC service components in the exit interview—weight measurement, blood pressure check-up, abdomen check-up, urine sample taken, blood sample taken, iron and folic acid and calcium tablets given—on that PMSMA day. We report coverage of quality ANC services by select sociodemographic and pregnancy

trimester, district and type of health facility. We did not include tetanus toxoid in this assessment as its administration in pregnant women is dependent on certain factors.[21] The tetanus toxoid coverage for last pregnancy in Bihar was reported at 89.5% in the most recent statewide assessment.[14] Though the PMSMA is primarily designed for women in second and third trimester of pregnancy, in reality, women in the first trimester also avail ANC services on PMSMA day. Therefore, we include these women in our analysis and present findings separately by pregnancy trimester; and report coverage of quality ANC services for the women in first trimester of pregnancy with and without considering abdomen check-up. We also report on whether the pregnant women were informed about the check-up provided to them and of the clinical findings by the health providers.

Using the facility observations of the ANC services flow, we compared the steps followed in the ANC service provision in each facility against what is recommended in the PMSMA guidelines to highlight the issues in process that could have implications on the quality of service provision.[22] STATA V.13.1 version was used for all analysis.

### Patient and public involvement statement

Patients were not involved in planning of this research study.

### RESULTS

A total of 961 pregnant women were identified for exit interviews of whom 861 (89.6%) were eligible for the study, and of whom 814 (94.5%) participated. Online supplemental table 1 documents the distribution of pregnant women based on the pregnancy trimester. Many of the pregnant women were in 20–24 years' age group (59.5%), 711 (87.3%) women were in their second or third trimester of pregnancy, and women belonging to other than forward caste (88%) accounted for most of the sample across the facilities (online supplemental table1).

### Coverage of each ANC component

The component-wise coverage of ANC services as reported in the exit interviews by the pregnant women is shown in online supplemental figure 1 and online supplemental table 2. Overall, weight check-up was reported by 98.4% (95% CI 97.3% to 99.1%) and blood pressure check-up by 97.1% (95% CI 95.7% to 98.2%) of the pregnant women (online supplemental figure 1). There was no significant difference in provision of these two components either by the pregnancy trimester or by the health facility (online supplemental figure 1). Though 80% of the women for whom weight was checked were also informed of the weight reading, less than one-third of them were informed about appropriateness of the weight gain as per the pregnancy trimester (table 1). Of the 790 women for whom blood pressure was checked, only 463 (58.7%) were informed of the blood pressure reading and of them only 223 (28.2%) were informed of the appropriateness

of the reading (table 1). There was considerable variation in women being informed about blood pressure by the pregnancy trimester (table 1).

The urine sample collection coverage was 46.3% (95% CI 42.9% to 49.7%) with significant variations seen by district, pregnancy trimester and facility (online supplemental figure 1). The blood sample collection coverage was 85.6% (95% CI 83.0% to 87.8%), with no significant variation by pregnancy trimester (online supplemental figure 1). Of the 697 women for whom blood sample was taken, only 235 (33.7%) were informed about the haemoglobin value, and almost none were informed about the blood sugar value (table 1).

The coverage of abdomen check-up was 62% (95% CI 58.6% to 65.3%) and significant variations were seen by the pregnancy trimester and health facility (online supplemental figure 1). Only 7.7% of women in second or third trimester were explained how to monitor baby's movements (table 1). Overall, 90.4% (95% CI 88.2% to 92.2%) of the pregnant women reported receiving IFA tablets and 92.1% (95% CI 90.1% to 93.8%) receiving calcium tablets, with no significant variations by pregnancy trimester or health facility (online supplemental figure 1). Of the 736 women who had received IFA tablets, only 193 (26.2%) were informed about the benefits and side effects of it. Only 114 (15.6%) women of the 750 women who had received calcium tablets were informed about the benefits of it (table 1).

### Quality ANC services

The coverage of quality ANC services irrespective of the pregnancy trimester was 30.4% (95% CI 27.3% to 33.7%), and was similar in both the districts (figure 1A) and by maternal age and caste (online supplemental table 2). This coverage was significantly lower for women in the first trimester of pregnancy (6.8%, 95% CI 3.3% to 13.6%) as compared with those in the second or third trimester of pregnancy (figure 1A). This coverage varied significantly between the facilities, and ranged from 3% to 66% in district 1 and 4.5%–83.1% in district 2 (figure 1A).

On not considering the abdomen check-up for women in the first trimester of pregnancy (figure 1B), quality ANC service coverage increased for them to 28.1% (95% CI 20.3% to 37.6%), and the pattern distribution by maternal age and caste was similar to that with inclusion of abdomen check-up (online supplemental table 2).

Among the 306 pregnant women in their third trimester of pregnancy (online supplemental table 1), 64 (20.9%) women had come for their first ANC visit, 95 (31.0%) for their second or third ANC visit and 147 (48%) for their fourth ANC visit or more. Not much variation was seen in the coverage of individual ANC components based on the number of ANC visit that they were in for (online supplemental figure 2). The coverage of quality ANC services was 34.4%, 26.3% and 36.1% for women who had come for their first, second or third, fourth visit or more ANC visit (online supplemental figure 2).

**Table 1** Distribution of pregnant women who participated in the exit interviews based on the component-wise antenatal care service provided on the given day and information provided to pregnant women regarding each of those components

| Service provided and women informed | All women N=814 (%) | Women in first trimester of pregnancy N=103 % (95% CI) | Women in second trimester of pregnancy N=405 % (95% CI) | Women in third trimester of pregnancy N=306 % (95% CI) |
|---|---|---|---|---|
| **Weight checked** | **801 (98.4)** | **99** <br> **96.1 (89.9 to 98.5)** | **402** <br> **99.3 (97.7 to 99.7)** | **300** <br> **98.0 (95.7 to 99.1)** |
| Informed the weight reading for whom weight was checked | 708 (88.4) | 82 <br> 82.8 (74.0 to 89.1) | 363 <br> 90.3 (86.9 to 92.8) | 263 <br> 87.7 (83.4 to 90.9) |
| Informed about appropriateness of weight gain by the stage of pregnancy for whom weight was checked | 246 (30.7) | 32 <br> 32.3 (23.8 to 42.2) | 125 <br> 31.1 (26.7 to 35.8) | 89 <br> 29.7 (24.7 to 35.1) |
| **Blood pressure checked** | **790 (97.0)** | **99** <br> **96.1 (90.1 to 98.5)** | **395** <br> **97.5 (95.5 to 98.7)** | **296** <br> **96.7 (94.0 to 98.2)** |
| Informed the blood pressure reading for whom blood pressure was checked | 463 (58.7) | 56 <br> 56.6 (46.6 to 66.0) | 237 <br> 60.1 (55.2 to 64.9) | 170 <br> 57.6 (51.9 to 63.2) |
| Informed about the appropriateness of blood pressure reading for whom blood pressure was checked | 223 (28.2) | 28 <br> 28.3 (20.2 to 38.0) | 114 <br> 28.9 (24.6 to 33.5) | 81 <br> 27.4 (22.6 to 32.7) |
| **Blood sample taken** | **697 (85.6)** | **94** <br> **91.3 (83.9 to 95.4)** | **346** <br> **85.4 (81.6 to 88.5)** | **257** <br> **83.9 (79.4 to 87.7)** |
| Informed that anaemia test will be done for whom blood sample was taken | 87 (12.5) | 13 <br> 13.8 (8.2 to 22.4) | 43 <br> 12.4 (9.3 to 16.3) | 31 <br> 12.1 (8.6 to 16.7) |
| Informed about the haemoglobin level for whom blood sample was taken | 235 (33.7) | 41 <br> 43.6 (33.9 to 53.8) | 123 <br> 35.5 (30.7 to 40.7) | 71 <br> 27.6 (22.5 to 33.4) |
| Informed about blood sugar level for whom blood sample was taken | 19 (2.7) | 1 <br> 1.1 (0.1 to 7.3) | 12 <br> 3.5 (2.0 to 6.0) | 6 <br> 2.3 (1.0 to 5.1) |
| **Abdomen checked** | **505 (62.0)** | **25** <br> **24.3 (16.9 to 33.5)** | **253** <br> **62.5 (57.6 to 67.1)** | **227** <br> **74.2 (69.0 to 78.8)** |
| Explained the process to monitor baby movements for whom abdomen was checked | 33 (7.7) | NA | 19 <br> 9.4 (6.0 to 14.2) | 15 <br> 6.6 (4.0 to 10.7) |
| **IFA tablets given** | **736 (90.4)** | **86** <br> **83.5 (74.9 to 89.5)** | **375** <br> **92.6 (89.6 to 94.8)** | **275** <br> **89.8 (85.9 to 92.8)** |
| Informed about the benefits and side effects to whom IFA tablets were given | 193 (26.2) | 19 <br> 23.5 (15.4 to 34.0) | 91 <br> 26.2 (21.8 to 31.1) | 83 <br> 31.6 (26.2 to 37.4) |
| **Calcium tablets given** | **750 (92.1)** | **88** <br> **85.4 (77.2 to 91.1)** | **383** <br> **94.6 (91.9 to 96.4)** | **279** <br> **91.2 (87.8 to 94.1)** |
| Informed about benefits to whom calcium tablets were given | 114 (15.6) | 8 <br> 9.3 (4.7 to 17.6) | 61 <br> 16.4 (13.0 to 20.6) | 45 <br> 16.6 (12.6 to 21.5) |

CI, confidence interval ; IFA, iron folic acid; NA, not applicable.

## Process of ANC services

The number and cadre of staff available for ANC services on the PMSMA day ranged from 6 to 13 across the various rounds of data collection in the two districts (online supplemental table 3). The average number of pregnant women examined per doctor (range 25.5–118), ANM

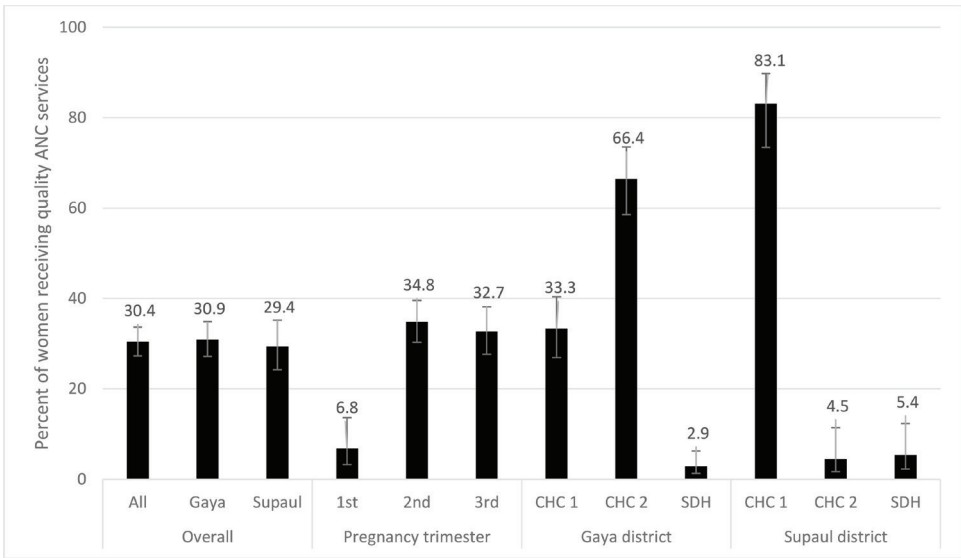

A

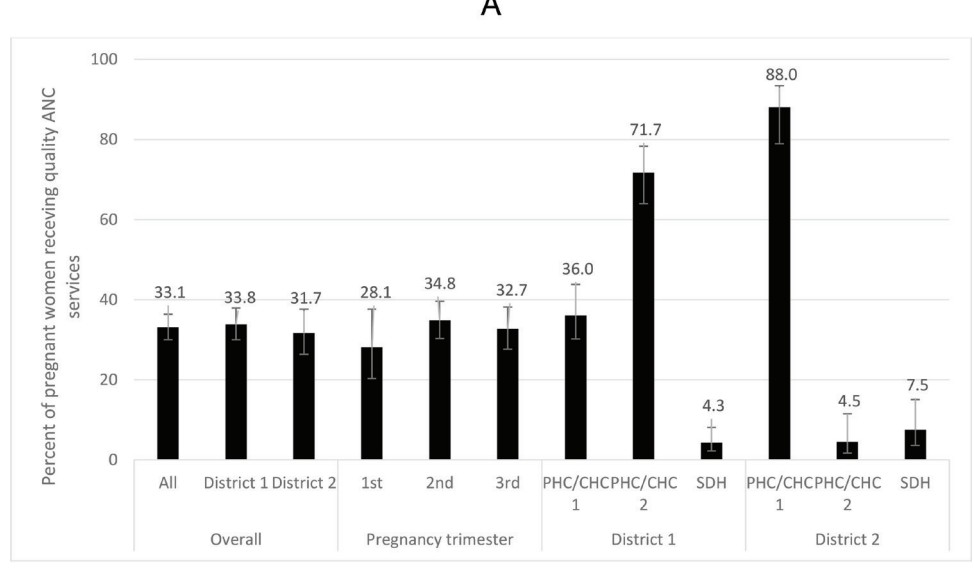

B

**Figure 1** Coverage of quality ANC services as reported by the pregnant women in exit interviews on the PMSMA day. Bars denote 95% CI. (A) Considering abdomen check-up for women in 1st trimester of pregnancy. (B) Not considering abdomen check-up for women in 1st trimester of pregnancy. ANC, antenatal care; CHC, community health centre; PMSMA, Pradhan Mantri Surakshit Matritva Abhiyan; SDH, subdistrict hospital.

(range 10.3–44.5) and lab technician (range 20.5–122) varied significantly between and within the facilities. The proportion of coverage of quality ANC services varied from 1.5% to 90.3%; a decrease in the proportion of quality ANC services was seen with increasing number of pregnant women seen per staff (p=0.230) and per ANM (p=0.121) but this was not statistically significant.

The process as per the PMSMA guidelines and what was observed in the facilities is shown in table 2. In all the facilities, pregnant women registered for ANC check-up at the registration counter and were given 'Out-patient department (OPD) slip' with a number. This OPD number was relevant only for that day and no facility had a system of tracking if a woman had visited for ANC services earlier. As per the PMSMA guidelines, step two is medical

check-up done by ANMs followed by laboratory investigations (step 3), collection of laboratory reports (step 4) and with this the pregnant women are seen by the doctor (step 5). However, such a process was not observed in any facility (table 2).

Doctor consultation (step 5) was done before the steps 2–4 in 4 of the six facilities. Of the 22 doctors who provided ANC services in these facilities across all the rounds, 14 (63.6%) were males; 11 (50%) were MBBS, 2 (9.1%) were MD, 4 (18.2%) were dental specialist, 5 (22.7%) were alternate medicine doctors. A total of 349 (65.9%) and 151 (58.9%) women in Gaya and Supaul districts were seen by a doctor, respectively. The doctors were observed mainly handing over prescription indicating blood and urine tests to be done, and the list of

**Table 2** Process of antenatal care (ANC) services as observed in the study facilities

| Steps suggested in the PMSMA guidelines | | Steps followed at the sampled facilities | | | | | |
|---|---|---|---|---|---|---|---|
| Step | Detail | Gaya district CHC 1 | Gaya district CHC 2 | Gaya district SDH | Supaul district CHC 1 | Supaul district CHC 2 | Supaul district SDH |
| 1 | Registration | Registration | Registration | Registration | Registration | Registration | Registration |
| 2 | ANC OPD by ANM/nurse | Doctor's consultation | Doctor's consultation | Doctor's consultation | Receipt of medicines | Laboratory investigations | ANC OPD by ANM/nurse |
| 3 | Laboratory investigations | Laboratory investigations | Laboratory investigations | Laboratory investigations | Laboratory investigations | Collection of laboratory reports | Doctor's consultation |
| 4 | Collection of laboratory reports | Collection of laboratory reports | Collection of laboratory reports | Collection of laboratory reports | Collection of laboratory reports | ANC OPD by ANM/nurse | Laboratory investigations |
| 5 | Doctor's consultation | ANC OPD by ANM/nurse | ANC OPD by ANM/nurse | ANC OPD by ANM/nurse | Doctor's consultation | Doctor's consultation | Collection of laboratory reports |
| 6 | Counselling | Receipt of medicines | Receipt of medicines | Receipt of medicines | ANC OPD by ANM/nurse | Receipt of medicines | Counselling |
| 7 | Receipt of medicines | | | | Counselling | | Receipt of medicines |
| 8 | Feedback and grievance redressal | | | | Feedback and grievance redressal | | Feedback and grievance redressal |

ANM, Auxiliary Nurse and Midwife; CHC, community health centre; OPD, outpatient department; PMSMA, Pradhan Mantri Surakshit Matritva Abhiyan; SDH, subdistrict hospital.

medicines to be provided to the pregnant women in all facilities except one where the doctor performed medical check-up.

Urine sample was taken only for 276 (50.3%) and 101 (38.1%) women in Gaya and Supaul districts, respectively. In some facilities, women were given new containers for urine sample collection and in some old containers, and all tests were done only using dipstick. No facility had a storage option available for urine samples. The laboratory technician noted the name and serial number for each pregnant woman in the laboratory register, and indicated the same number on the container. Women were asked to go to the toilet with the container and dipstick, to dip the dipstick one inch in the urine, and bring it back to the technician. No laboratory technician was seen handling the dipstick; women were asked to hold it up and the technician recorded the reading from a distance. The field team noticed a make-shift toilet (online supplemental figure 3) in one facility, and the toilet in one facility was on another floor than the ANC OPD. Pregnant women were seen hesitating in using both of these toilets, and some did not take the urine test even after being given the container. In the remaining facilities, pregnant women had to use a toilet elsewhere, which was not close to the laboratory. Some were noticed being embarrassed at carrying the container and dipstick, as the other people present in the facility could see them. Blood sample was taken only for 437 (79.6%) and 260 (98.1%)

women in Gaya and Supaul districts, respectively. Venous blood was drawn for 250 (35.9%) and the rest was finger prick method. The laboratory technicians were not seen wearing gloves every time they took a blood sample. Blood samples were discarded immediately after recording of the haemoglobin value in all facilities.

Almost all the check-up that the pregnant women were offered was done by the ANMs (Auxiliary Nurse and Midwife) in all the facilities. None of the clinical findings by the ANMs were checked by the doctor, as the women were made to see the doctor before any check-up was done. In all facilities, the ANMs entered the clinical data in the PMSMA register next day and not immediately after examining the women. The counselling services (step 6) and feedback redressal (step 7) were seen only in two of the six facilities. Only 38 (18.3%) and 11 (11.8%) of the women in SDH of Gaya and Supaul districts reported meeting a counsellor, respectively.

## DISCUSSION

In our understanding, this is one of the first studies undertaken in a health facility in a low/middle-income country setting that provides a systematic documentation of quality of ANC services considering all the ANC components through exit survey, along with documentation of the process of ANC service provision. Only one-third of the pregnant women had received quality ANC

services under the PMSMA, and even less were informed about the services that they had received. The wide variations in the implementation of PMSMA at the facilities, and the task shifting for service provision will have to be addressed urgently to improve the quality of ANC service provision.

Two components that resulted most in poor quality of ANC services were urine examination and abdomen check-up. This study has highlighted that good clinical laboratory practice guidelines were not followed for urine examination in any of the facilities.[23] Notably, though the coverage of blood sample examination was relatively higher than that for urine, however, only blood sugar and anaemia levels were recorded and none of the other tests recommended in pregnancy were performed. These findings suggest improving the laboratory readiness is imperative to improve quality of ANC services.[24] The laboratory infrastructure and related processes, and the laboratory technicians as human resources for health have not received the necessary attention in provision of health services until recently in India.[25–27] Laboratory services have recently received unprecedented attention in India due to COVID-19, and it would be important to sustain this and expand it to routine pregnancy laboratory tests as well.[28 29]

The Indian ANC guidelines recommend abdomen check-up from second pregnancy trimester onwards to monitor pregnancy progression, fetal well-being and its position.[24] We have previously reported from this population that breech position is a significant risk factor for early neonatal deaths and stillbirths, however, breech position of the baby is not known to most women before delivery as abdomen check-up is either not done or they are not informed.[30 31] Abnormal fetal movements can be used to predict adverse neonatal outcomes,[32–34] however, only 1 in 12 women who had received abdomen check-up was informed about monitoring of baby movements in this study.

Weight and blood pressure measurements, and provision of IFA and calcium tablets showed consistently high coverage across all the facilities irrespective of the pregnancy trimester. However, less than one-third of the pregnant women were informed about the findings or implications of these for pregnancy. This coupled with almost non-existence of counselling is a major concern. Low birth weight is reported to be the largest contributor to child malnutrition disability-adjusted-life years in India,[35] and birth weight is an intergenerational issue dependent on an interplay of various factors, including maternal undernutrition and intrauterine growth. Also, anaemia increases the risk of adverse birth outcomes, and the high burden of anaemia in Indian women has not declined since 1990.[35] With very poor communication with the women and almost no counselling, pregnant women are not empowered to take correct steps for a positive pregnancy experience and to prevent adverse birth outcomes.[36]

Substantial inequities in coverage of quality ANC services by socioeconomic status have been highlighted in some

previous assessments.[1 37] As this study was carried out only in the public sector facilities, we are unable to comment on inequities in general. However, it is important to note that the coverage of quality ANC services in our study was similar irrespective of the caste of women, which is a surrogate measure of socioeconomic status in rural India. The new WHO ANC model recommends that women attend a minimum of eight ANC visits.[36] However, with the significantly poor levels of quality ANC services as documented in this study and other assessments, focusing only on increasing the number of visits is unlikely to produce the desired maternal and neonatal health outcomes.[38] With coverage of quality ANC services at 30% on PMSMA day, the effort to improve quality of ANC can go only thus far unless the quality of ANC is explicitly tracked and monitored through standard indicators both at the health system and community levels.[38–40] Though the India Newborn Action Plan recommends monitoring of percentage of pregnant women who received full ANC but the definition of full ANC is not provided.[41]

The varied availability of staff on PMSMA day between and within the facilities over the various rounds of data collection in this study, and the substantial variations found in the steps of PMSMA implementation at the facilities, are of concern with regard to the provision of quality services. Despite the premise of PMSMA day being that every pregnant woman in India is examined by a physician and appropriately investigated at least once during the PMSMA,[12] almost all the check-up in our study was provided by ANMs in all the facilities. Notably, the doctor's consultation did not involve any examination of the pregnant women in all but one facility, even though the availability of doctors was not an issue in any facility. These findings indicate task shifting which is neither in line with the PMSMA guidelines[12] nor with the task shifting recommendations to maintain the quality of services.[42] Importantly, the programme implementation needs to address and account for such task shifting to address the poor quality of ANC services. The wide variation in coverage of quality also indicates the scope for improvement. Raising the performance of all health facilities to the level of best performance should be feasible with more in-depth understanding of implementation issues at the facility level, which would lead to significant overall gains in quality and ensure that pregnant women receive quality services irrespective of the facility they access for these services.[7]

There are some limitations of the study. It was conducted in six facilities only, which could be considered as a limitation to generalise these findings. It is important to note that concerns have also been raised with the quality of maternal services in the private sector as well, and it would be important to assess quality of ANC services for these facilities.[43–46] Furthermore, we were able to conduct data collection for only one round instead of three in one district. However, it is important to note that the study findings corroborate with the previous findings of poor coverage of quality ANC services in household

surveys in the state. We excluded 10% of women because of non-availability of a phone for contacting her later for follow-up assessment. However, this proportion is small to have any significant impact on the study findings.

There are several strengths of this study. There is almost no recall bias in the information reported by the respondents on the ANC services received, as this was an exit survey. The nature of this study which includes patient exit surveys immediately after ANC check-up, documentation of ANC services components beyond what is previously reported, documentation of what women are informed, and the process of ANC services are strengths of this undertaking.

## CONCLUSION

This study highlights grossly inadequate quality of ANC services in public sector health facilities. The findings suggest that in order to provide ANC services as envisaged under the PMSMA, there is an urgent need to cultivate quality in ANC service provision at the public health facilities; to train doctors, ANMs and laboratory technicians to communicate with the pregnant women for a positive pregnancy experience and to address complications; and to build clinical and technical capacity and supervision for health providers to follow the standard guidelines for provision of quality ANC services.

**Acknowledgements** The authors acknowledge the contributions of Sibin George and S Siva Prasad from Public Health Foundation of India, and Asif Iqbal and Vipul Singhal from the Oxford Policy Management, India for data collection and data management.

**Contributors** RD and AK had full access to data in the study, take full responsibility for the integrity of data and accuracy of the data analysis, and had final responsibility for the decision to submit for publication; RD, AK and LD conceptualised the study; RD guided the data analysis and drafted the manuscript; MM and AK performed data analysis; MA guided data collection; MA, DB and PN contributed to data analysis and interpretation; LD contributed to drafting of the manuscript and interpretation; all authors approved the final manuscript. RD is responsible for the overall content as the guarantor.

**Funding** Bill & Melinda Gates Foundation; grant number INV-007989.

**Competing interests** DB and PN are employees of the Bill & Melinda Gates Foundation, India Country Office, New Delhi. The other authors have no conflict of interest.

**Patient and public involvement** Patients and/or the public were not involved in the design, or conduct, or reporting, or dissemination plans of this research.

**Patient consent for publication** Consent obtained directly from patient(s).

**Ethics approval** This study was approved by the Ethics Committee of the Public Health Foundation of India (TRC-IEC 410.1/19). Participants gave informed consent to participate in the study before taking part.

**Provenance and peer review** Not commissioned; externally peer reviewed.

**Data availability statement** All data relevant to the study are included in the article or uploaded as online supplemental information.

**ORCID iD**
Rakhi Dandona http://orcid.org/0000-0003-0926-788X

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
