## [Reviewer comments · BMJ Open]

ARTICLE DETAILS

TITLE (PROVISIONAL)	Assessment of quality of antenatal care services in public-sector facilities in India
AUTHORS	Dandona, Rakhi; Majumder, Moutushi; Akbar, Md.; Bhattacharya, Debarshi; Nanda, Priya; KUMAR, ANIL; Dandona, Lalit

VERSION 1 – REVIEW

REVIEWER	Samiksha Singh Indian Institute of Public Health, Hyderabad
REVIEW RETURNED	29-Aug-2022

GENERAL COMMENTS	Dear Author, It is a well written manuscript. I have only two comments and require clarifications in the text. 1. about 10% women who did not have cellphone were excluded. Why couldn't we plan to track them through the ANC checkup? What were the socio-demographic characteristics of these women and how excluding them would have affected your results.2. I believe that in overwhelmed Indian health facilities, the overcrowding and time constraints negatively influence the quality of service provided. I have also observed that during these days the centers are busier than usual. Did you assess or try to assess and discuss the extent of this problem, and what are the solutions to improve the quality of ANC services amidst many other gaps you identified. Kindly add in discussion. Best wishes to authors.
---

REVIEWER	Martina Mchenga Stellenbosch University
REVIEW RETURNED	15-Sep-2022

GENERAL COMMENTS	I would really like to recommend the authors for the work done so far and I think it is an important topic. However, I still feel like the methods were inadequate and do not do a great job at answering the research question. My main suggestion would therefore be for the authors to revise their methods and make them more clear. 1. The authors have only reported on the exit interviews but they did not clearly mention whether they also did direct observations using the PMSMA guidelines? It could help if that is made very clear in the methods section.2. This question relates to the first. I am trying to understand whether this was more of a qualitative study and narrative and that the exit interviews were done to supplement the information which was collected? The authors mentioned that they compared the steps
---

	followed in ANC against those PMSMA guidelines, how did they do that? Did they a third party expert observing the provider? 3. How did the authors end up identifying 961? What statistical methods were used to come up with that number? 4. Table 1, where are the statistics based from? Is it from client exit interviews? What is the source of the data? 5. The authors must also specify which component of quality of care they are focusing and make sure to make that clear in their presentation of results as well interpretation. 6. The issue on task shifting to nurses and the authors argument that it may affect quality of care. Although that maybe try for specialised care, evidence has shown that when it comes to primary health care which ANC is one of the components, nurses do a much better job than doctors. Could the authors conduct some regressions to check and compare whether the quality of care varied by the type of provider? In general, I think the paper can benefit from revising the methods and how they help in answering the research question.
--	--

REVIEWER	Juraci Cesar Universidade Federal do Rio Grande, Population & Health
REVIEW RETURNED	17-Sep-2022

GENERAL COMMENTS	General comments:  - This is a cross-sectional study with a prospective collection of information among pregnant women attended at six primary health care units in different districts. This data collection consisted of certifying the care received for several indicators after prenatal visits in health services. Losses were identified but were few and similarly distributed in the studied groups. The most significant differences concerning coverage assessed resulted from pregnant women being approached in the different trimesters of pregnancy. The more advanced the pregnancy, the greater the coverage achieved. This study can contribute to improving the implementation of measures to enhance the quality of prenatal coverage in the studied location and similar settings. Specific comments: Abstract:  - The sentence, "The inadequate quality of antenatal care (ANC) services is mainly reported from population surveys in developing settings." should be deleted because it is not an objective: - Considering that the term quality means "the degree of excellence of something", why not classify pregnant women who received all these services as having had adequate ANC? As presented, the "quality of ANC services" is unclear and vague. - I suggest mentioning coverage (with CI) in the second and third trimesters of pregnancy in this section. Furthermore, it will be relevant to provide any data to support that information sharing with pregnant women after check-up is poor. - The conclusion presented does not meet and does not seek to respond to the study's objectives. These two points should be better
--

	tighter. Background: - The use of only two paragraphs and the lack of connection between its sentences hinders a better appreciation of this section. The investigation question does not seem to be properly stated. - It is necessary to value more how much this manuscript can contribute to the gap pointed out: "In addition to documenting the significantly inadequate quality of ANC services, the study documents the poor information sharing with the pregnant women based on the check-up, and deviations from the prescribed process of ANC service provision." Methods: - It is challenging to understand this section. The order in which the topics are presented does not follow the expected sequence. The same paragraph addresses different issues. Moreover, there is no reasonable connection between the sentences and paragraphs. Results: - This section is too long. It practically describes all the results presented in the tables. The text could be more selective, presenting only the most relevant results. Discussion: - This section could be better connected with the study objectives. I suggest including a paragraph explicitly responding to what was sought at the end of this section.
--	---

VERSION 1 – AUTHOR RESPONSE

Reviewer 1

Dr. Samiksha Singh, Indian Institute of Public Health, Hyderabad

It is a well written manuscript. I have only two comments and require clarifications in the text.

1. About 10% women who did not have cellphone were excluded. Why couldn't we plan to track them through the ANC checkup? What were the socio-demographic characteristics of these women and how excluding them would have affected your results.

It was not possible to track all the recruited women through to the next ANC check-up at the facility as it is not necessary that all them would come for the next ANC visit on the same day and to the same facility. Such a study design would have had major implications for the study resources and logistics of data collection. Therefore, the follow-up interview with these women was planned over phone and not in person. Around 10% of the women were excluded, and we did not collect their socio-demography details. Given that only 10% women were excluded, the proportion is not significant to affect the study findings. This is shown as a limitation (lines 343-345).

2. I believe that in overwhelmed Indian health facilities, the overcrowding and time constraints negatively influence the quality of service provided. I have also observed that during these days the centers are busier than usual. Did you assess or try to assess and discuss the extent of this problem, and what are the solutions to improve the quality of ANC services amidst many other gaps you identified. Kindly add in discussion.

Thank you for this observation. Supplementary Table 3 shows the number of pregnant women seen per staff and per type of health provider. We have now added the coverage of quality ANC services for each round in this Table. We did not find a statistically significant association between the number of pregnant women per staff/ANM with the quality of ANC services though a decreasing trend in quality was seen with increasing number of pregnant women (lines 220-223).

We cannot comment on the overall patient load in these facilities and quality of services as that was beyond the scope of this study.

Reviewer 2

Dr. Martina Mchenga, Stellenbosch University

I would really like to recommend the authors for the work done so far and I think it is an important topic. However, I still feel like the methods were inadequate and do not do a great job at answering the research question. My main suggestion would therefore be for the authors to revise their methods and make them more clear.

Thank you. We have attempted to make the methods clearer.

1. The authors have only reported on the exit interviews but they did not clearly mention whether they also did direct observations using the PMSMA guidelines? It could help if that is made very clear in the methods section.

Thank you. We have conducted exit surveys and direct observations. We have made some changes in the methods structure to convey this more clearly.

2. This question relates to the first. I am trying to understand whether this was more of a qualitative study and narrative and that the exit interviews were done to supplement the information which was collected? The authors mentioned that they compared the steps followed in ANC against those PMSMA guidelines, how did they do that? Did they a third party expert observing the provider?

The exit interviews and observations were two separate components. We have made some changes in the methods structure to convey this more clearly.

3. How did the authors end up identifying 961? What statistical methods were used to come up with that number?

Lines 99-100 – We have stated in the methods that the desired sample size was 1,200 pregnant women with the aim to get 200 pregnant women across each sampled health facility.

4. Table 1, where are the statistics based from? Is it from client exit interviews? What is the source of the data?

These are from the exit interviews. This is now stated in the Table 1 header.

5. The authors must also specify which component of quality of care they are focusing and make sure to make that clear in their presentation of results as well interpretation.

This is already stated in methods lines 137-140.

6. The issue on task shifting to nurses and the authors argument that it may affect quality of care. Although that maybe try for specialised care, evidence has shown that when it comes to primary health care which ANC is one of the components, nurses do a much better job than doctors. Could the authors conduct some regressions to check and compare whether the quality of care varied by the type of provider?

The primary purpose of the PMSMA program is that if every pregnant woman in India is examined by a physician and appropriately investigated at least once during the PMSMA and then appropriately followed up, the process can result in reduction in the number of maternal and neonatal deaths in the country (lines 65-67). The program defines the tasks for each health provider. The point we are making here is that the task shifting documented in our study is not allowed for by the program. As stated in lines 240-242, the doctors were observed mainly handing over prescription indicating blood and urine tests to be done, and the list of medicines to be provided to the pregnant women in all facilities except one where the doctor performed medical check-up. This is against the primary purpose of the program which is built upon

examination of the pregnant woman by a physician. This study is not designed to comment on whether the nurses/ANMs do better job than the doctors, but it documents that they do the job that is to be done by the doctors.

7. In general, I think the paper can benefit from revising the methods and how they help in answering the research question.

Thank you. We have introduced sub-sections in the methods to make it more reader friendly, and have re-worked on some text to improve clarity.

Reviewer 3

Dr. Juraci Cesar, Universidade Federal do Rio Grande

General comments:

This is a cross-sectional study with a prospective collection of information among pregnant women attended at six primary health care units in different districts. This data collection consisted of certifying the care received for several indicators after prenatal visits in health services. Losses were identified but were few and similarly distributed in the studied groups. The most significant differences concerning coverage assessed resulted from pregnant women being approached in the different trimesters of pregnancy. The more advanced the pregnancy, the greater the coverage achieved. This study can contribute to improving the implementation of measures to enhance the quality of prenatal coverage in the studied location and similar settings.

Thank you.

Abstract:

1. The sentence, "The inadequate quality of antenatal care (ANC) services is mainly reported from population surveys in developing settings." should be deleted because it is not an objective:

This sentence is now deleted.

2. Considering that the term quality means "the degree of excellence of something", why not classify pregnant women who received all these services as having had adequate ANC? As presented, the "quality of ANC services" is unclear and vague.

Lines 25-27 - Quality ANC services is defined for woman. If a woman received all of these services in that visit - weight, blood pressure and abdomen check, urine and blood sample taken, and were given iron and folic acid and calcium tablets – she was considered to have received quality ANC services.

3. I suggest mentioning coverage (with CI) in the second and third trimesters of pregnancy in this section. Furthermore, it will be relevant to provide any data to support that information sharing with pregnant women after check-up is poor.

Coverage in 2nd and 3rd trimester pregnancy is now added in line 33. Adding more data to support that information sharing with pregnant women after check-up is poor will add many more words, which will take abstract beyond the recommended word count by BMJ Open.

4. The conclusion presented does not meet and does not seek to respond to the study's objectives. These two points should be better tighter.

Lines 41-42 - We have re-worked on this sentence.

Background:

5. The use of only two paragraphs and the lack of connection between its sentences hinders a better appreciation of this section. The investigation question does not seem to be properly stated.

Lines 60-67 - We have modified the last part of paragraph 1 and the first part of paragraph 2 to make a better connection between the two paragraphs.

6. It is necessary to value more how much this manuscript can contribute to the gap pointed out: "In addition to documenting the significantly inadequate quality of ANC services, the study documents the poor information sharing with the pregnant women based on the check-up, and deviations from the prescribed process of ANC service provision."

Thank you. We have now re-written lines 75-78 to convey what the manuscript contributes and to better link it to the overall aim.

Methods:

7. It is challenging to understand this section. The order in which the topics are presented does not follow the expected sequence. The same paragraph addresses different issues. Moreover, there is no reasonable connection between the sentences and paragraphs.

Thank you. We have now organised the methods section with sub-headings to make it more clear and sequential. We have also re-organised some text for more clarity.

Results:

8. This section is too long. It practically describes all the results presented in the tables. The text could be more selective, presenting only the most relevant results.

Given the denominator for each ANC component varies based on the number of women offered that service, it is important to state this for each component as the subsequent proportions are from these specific denominators. This is adding to the text in this section. However, we have attempted to shorten the results from exit interviews.

Discussion:

9. This section could be better connected with the study objectives. I suggest including a paragraph explicitly responding to what was sought at the end of this section.

We have reorganised the text in this section to improve the connect with the study objectives. Now, the exit interview results are discussed first followed by the results from the observations. Also as suggested by the Editor, we have now moved the last paragraph to "conclusion" section (lines 353-360), and have expanded it to clearly connect with the study findings.

VERSION 2 – REVIEW

REVIEWER	Martina Mchenga Stellenbosch University
REVIEW RETURNED	11-Oct-2022

GENERAL COMMENTS	I still feel that the paper is lacking and should not published in its current state.  1. The abstract is lacking a background section 2. Statement on line 60-62, the better justification would be retrospective data is prone to recall errors and therefore can be biased. 3. I still think the paper can benefit from having proof read by an independent English editor. 4. The authors indicated that 10% of the women who were loss to follow up due to lack of mobile phones is not a significant number. But if you add that to the women who did not consent to participate in the study, it may have significant impact on the outcome of the analysis.
--

	5. Line 75, need to correct punctuation, pretty much through out the paper. For example, mortality ratio of 230.[15 16] should be "mortality ratio of 230 [15 16]." The punctuation mark should come after the closed bracket.
--	--

REVIEWER	Juraci Cesar Universidade Federal do Rio Grande, Population & Health
REVIEW RETURNED	08-Oct-2022

GENERAL COMMENTS	Dear Dr. Rakhi Dandona et al. Congratulations on this important piece of work. I am sure it will be relevant to improve antenatal care in different countries. I believe that the manuscript now has all the conditions to be published in the BMJ Open. Yours sincerely, Juraci A. Cesar Associate Professor Federal University of Rio Grande, Brazil.
--

VERSION 2 – AUTHOR RESPONSE

Reviewer: 3

Thank you.

Reviewer: 2

1. The abstract is lacking a background section.

The BMJ Open guidelines for abstract do not require a background section.

2. Statement on line 60-62, the better justification would be retrospective data is prone to recall errors and therefore can be biased.

Thank you. The references with this statement are already cited (now Lines 62-64). Of note, we have not indicated that the retrospective data are biased but have indicated that these are not nuanced enough to offer an understanding of what happens in a particular ANC visit with a health provider.

3. I still think the paper can benefit from having proof read by an independent English editor.

Thank you. We leave this to the Journal to decide.

4. The authors indicated that 10% of the women who were loss to follow up due to lack of mobile phones is not a significant number. But if you add that to the women who did not consent to participate in the study, it may have significant impact on the outcome of the analysis.

We disagree. The findings are not significantly impacted by this exclusion based on our understanding of our population and local context. We have already listed this as a limitation.

5. Line 75, need to correct punctuation, pretty much through out the paper. For example, mortality ratio of 230.[15 16] should be "mortality ratio of 230 [15 16]." The punctuation mark should come after the closed bracket.

This is so only where there is a reference, and it is automated by the Endnote and not done manually. This reference style was changed at the request of the journal with the previous revision.

VERSION 3 – REVIEW

REVIEWER	Martina Mchenga Stellenbosch University
REVIEW RETURNED	10-Nov-2022
GENERAL COMMENTS	The reviewer completed the checklist but made no further comments.